# Sports Injuries of a Portuguese Professional Football Team during Three Consecutive Seasons

**DOI:** 10.3390/ijerph191912582

**Published:** 2022-10-02

**Authors:** Francisco Martins, Cíntia França, Adilson Marques, Beatriz Iglésias, Hugo Sarmento, Ricardo Henriques, Andreas Ihle, Helder Lopes, Rui T. Ornelas, Élvio Rúbio Gouveia

**Affiliations:** 1Department of Physical Education and Sport, University of Madeira, 9020-105 Funchal, Portugal; 2Laboratory of Robotics and Engineering Systems, Interactive Technologies Institute, 9020-105 Funchal, Portugal; 3CIPER, Faculty of Human Kinetics, University of Lisbon, 1499-002 Lisbon, Portugal; 4ISAMB, Faculty of Medicine, University of Lisbon, 1649-020 Lisbon, Portugal; 5Faculty of Human Kinetics, University of Lisbon, 1499-002 Lisbon, Portugal; 6Research Unit for Sport and Physical Activity (CIDAF), Faculty of Sports Sciences and Physical Education, University of Coimbra, 2004-504 Coimbra, Portugal; 7Marítimo da Madeira—Futebol, SAD, 9020-208 Funchal, Portugal; 8Department of Psychology, University of Geneva, 1205 Geneva, Switzerland; 9Center for the Interdisciplinary Study of Gerontology and Vulnerability, University of Geneva, 1205 Geneva, Switzerland; 10Swiss National Centre of Competence in Research LIVES—Overcoming Vulnerability: Life Course Perspectives, 1015 Lausanne, Switzerland

**Keywords:** football injuries, risk factors, epidemiology, sports monitoring, injury prevention

## Abstract

Professional football players are exposed to high injury risk due to the physical demands of this sport. The purpose of this study was to characterize the injuries of a professional football team in the First Portuguese League over three consecutive sports seasons. Seventy-one male professional football players in the First Portuguese Football League were followed throughout the sports seasons of 2019/2020, 2020/2021, and 2021/2022. In total, 84 injuries were recorded. Each player missed an average of 16.6 days per injury. Lower limbs were massively affected by injuries across all three seasons, mainly with muscular injuries in the quadriceps and hamstrings and sprains in the tibiotarsal structure. Overall, the injury incidence was considerably higher in matches than in training. The two times of the season that proved most conducive to injuries were the months of July and January. Our results emphasize the importance of monitoring sports performance, including injury occurrence, and assisting in identifying risk factors in professional football. Designing individualized training programs and optimizing prevention and recovery protocols are crucial for maximizing this global process.

## 1. Introduction

Football is known for its fast-paced and intensive activities from a professional standpoint [1,2], which certainly contributes to a higher injury risk during the season [3]. Sports injuries not only negatively affect the players’ careers but also the performance of their teams and clubs [3,4,5,6,7,8,9]. It is one of the most significant issues that professional football players must face [10]. Consequently, football clubs have increased their interest in having feasible, reliable, fast, and accessible information that can help sustain coaches, trainers, and medical staff decisions about the physical condition of their players [11].

The main identified factor that prevents professional football players from not being able to train and play during the football season is sports injuries. Indeed, there are several studies that address the topic of sports injuries in football, mainly professionally [3,5,6,8,10,12,13,14,15,16,17,18,19,20,21,22,23,24,25,26,27,28,29,30], but also with young football athletes [4,7,31,32,33,34]. The main areas of intervention in these numerous studies are the frequency and incidence of injuries in football, the comparison between various sports seasons and different competitive levels, the prediction of the injury risk in soccer, and injury prevention programs. In particular, a study in Chile and another in Spain performed the same procedure of monitoring a professional football team over three consecutive sports seasons [3,14]. Other studies present higher temporal follow-up of professional football teams or leagues [6,13,15,16,29].

The previous literature has reported that injuries are responsible for 49% of match unavailability and 60% of training unavailability [12]. As this is becoming an urgent issue, the activities performed in training sessions must reflect the demands of competition, focusing on the development of players’ performance, which includes injury prevention strategies [35,36,37].

An injury is defined as an event that occurs during a scheduled training session or match, resulting in an absence from the next training session or match [13]. The sectorial positions most affected by injuries are defense and forward [3,14,31]. In contrast, goalkeepers frequently present the lowest risk of injury [31]. Past studies on football injuries have described the lower limbs as the most affected body zone [3,14,15,16,17,18,31], particularly for muscle injuries in the thigh area, the quadriceps, and the groin [3,16,19]. Several investigations stated that 20% to 37% of all time-loss injuries at a professional level are muscular injuries [13,18,20,21,22]. Naturally, advanced age contributes to a higher injury exposure [15,18], and knee ligament injuries typically cause the most severity and time away from training and competition [23].

A study conducted over three consecutive seasons concluded that a professional football team could expect 1.5 injuries per number of players on their roster [3]. Regarding the most frequent mechanism of injury, there is some balance in the results presented in the literature, with some studies referring to a predominance of traumatic injuries [3] and others the prevalence of overload injuries [17]. On average, a player misses training and competition after contracting an injury for 7 to 8 days [3,17]. The injury recurrence rate ranges from 8% to 22% [15,23,24].

Collecting and evaluating the data on injury risk factors that may contribute to the incidence of sports injuries is crucial. This knowledge will improve injury prevention and treatment and, consequently, teams’ performance [14,17]. Therefore, this study aimed to describe and characterize the injuries of a professional football team in the First Portuguese League over three consecutive sports seasons. As far as we know, this investigation is the first to explore injury characterization over three seasons in this professional football league. It was hypothesized that the sporting season in which this professional team achieved the worst classificatory result in the First Portuguese League is the year in which the team has contracted a higher number of injuries and that there is a relationship between the variables that characterize sports injuries and the team’s sporting performance. The results were mostly in line with the main research hypothesis outlined in this study.

## 2. Materials and Methods

### 2.1. Study Design

This descriptive, observational study was carried out in a professional men’s football team competing in the First Portuguese League during the sporting seasons of 2019/2020, 2020/2021, and 2021/2022. It is essential to note that this professional team presented six head coaches over the three seasons under review. Interestingly, the season with the most injuries (2020/2021) was also the one with the most coaches (i.e., three in total). Furthermore, the 2019/2020 season was marked by the COVID-19 pandemic, which caused the First Portuguese League competition to be interrupted from 8 March (round 24) to 3 June (round 25), making it an atypical season. All the applied procedures were approved by the Faculty of Human Kinetics Ethics Committee (CEIFMH No. 34/2021). The research was conducted according to the principles of the Declaration of Helsinki, and all the players signed informed consent for participation in this study. In addition, all the participants were treated ethically according to the American Psychological Association code of ethics. The medical department performed daily injury records during the season, including training and competitive moments. The type, frequency, zone, specific location, mechanism, severity, recurrence, occurrence, and moment of the injury were recorded.

### 2.2. Participants

In total, 71 male professional football players (25.7 ± 3.4 years old; 181.6 ± 6.5 cm; 77 ± 7.2 kg) participated in this study, comprising 8 goalkeepers (11.3%), 20 defenders (28.2%), 17 midfielders (23.9%), and 26 forwards (36.6%). Fifty-two players had the right lower limb as dominant (73.2%), and nineteen had the left lower limb as dominant (26.8%). All the players who represented this team were included in the study, even those who reinforced or left the team during the seasons. The players injured at the end of a season were followed until the end of their recovery period.

### 2.3. Procedures

Regarding the variables under analysis, the frequency of injuries by age corresponds to the number of injuries accounted for in each age group. The frequency of injuries by sectorial position is defined by the number of injuries that goalkeepers, defenders, midfielders, and forwards contracted. The type, zone, and specific location of the injury are complementary variables that identify the part of the body that suffered structural and/or functional changes. The mechanism of injury is intended to understand if the injury was traumatic, or if it was contracted by overload. The severity of the injury considers the period, in days, from the athlete’s stoppage until resuming field work with the consent of the clinical department. The injury occurrence is characterized by the work session (training or competition) that the athlete was performing when the injury was contracted. The exposure time of the athletes throughout the seasons was collected using a 10 Hz GPS device (EVO, Catapult, Melbourne, Australia) during each training session and an official match. The GPS device was put in a skin-tight bag in the thoracic region between the scapulae. The injury incidence was calculated as the number of injuries contracted during a sporting activity divided by 1000 h of exposure time, multiplied by the exposure time collected with the GPS device in game and training situations. Laterality corresponds to whether the player contracted the injury in his dominant or non-dominant limb. Finally, an injury was marked as recurrent when a player was injured in the same place and type where he was previously affected by an injury.

Throughout the three seasons analyzed, the professional soccer team of Club Sport Marítimo always presented the same physical trainer responsible for the area of injury prevention and training load control. This professional in the sports area was responsible for exposing this team to a preventive injury workload, mainly based on mobility, myofascial, proprioceptive, plyometric, eccentric strength, bilateral and unilateral strength, isometric, and dynamic core exercises. These sessions varied between the field and the gym, having some differences in the number of sessions, series, and weekly moments throughout the three seasons analyzed, mainly depending on the vision of the coach who was in charge of the team in each sports moment. Even so, the athletes performed such preventive exercises, on average, 2 sets with 4 to 8 repetitions, with durations between 15 and 25 min.

The main significant difference between sports seasons was related to the equipment and techniques used daily and weekly to control the players’ physical and psychological states. In the 2019/2020 season, the players started 3 out of 5 training sessions with a thermographic evaluation. In all field training sessions, a GPS was used. On specific days of the week, they performed ice baths and sports massages to optimize their general body condition. The care and monitoring of processes such as hydration, adequate hours of sleep, supplementation, and nutrition were central to the optimization of this process. For the 2020/2021 season, all the preventive work and control of the players’ conditions were improved by adding to all the aspects already carried out in the previous season a Wellness evaluation before each training session and an RPE evaluation at the end of each training session. In addition, a weekly evaluation was scheduled, always performed two days before the day of the game, with precise evaluation tools such as optojump (countermovement jump and squat jump), a hip adductor squeeze test, and InBody 770. These weekly evaluations allowed both the players and the coaching staff to have more precise and individualized data about the physical and psychological abilities of the players on a weekly basis, having the possibility to adjust the type, intensity, and load of the training sessions, aiming to maximize the sporting performance of each player. This was the first sports season in which weekly evaluations started to be implemented. This process required some time until its full implementation at Club Sport Marítimo. Consequently, the pre-season phase was the one in which the injury preventive processes were less applied due to the difficult logistics of implementing weekly evaluations. Although in this season, there were three coaches ahead of this professional team, with changes inherent to the whole process, the preventive work and evaluations were always kept by the club. Finally, in the 2021/2022 season, this whole process was optimized and followed from the pre-season to the last competitive week of this professional team. Although different coaches have different methodologies and visions regarding training, this work consisting of prevention, evaluation, and daily follow-up of the players who were part of this professional team’s roster throughout the three analyzed seasons was fundamental. The biological, physiological, and psychological conditions are also factors that influence the players’ performance. Due to the multidimensional impact of the factors to which the players are exposed on a daily basis, these data can increase the awareness of sports agents and coaches to consider prevention assessment as an essential tool in monitoring the players and implementing preventive programs and measures during sports seasons.

### 2.4. Statistical Analysis

Descriptive statistics were used to summarize the data collected. Absolute values present the number of football players and the total number of injuries. The demographic data of the participants are presented by mean and standard deviation. The frequency of the injuries by age, sectorial position, type, zone, specific location, laterality, mechanism, severity, recurrence, and occurrence are represented by absolute values and their percentages. All the analyses were performed using the IBM SPSS Statistics software 26.0 (SPSS Inc., Chicago, IL, USA).

## 3. Results

Table 1 displays the team’s competitive pathway during the three seasons under analysis, while Table 2 presents the respective injury report. Based on this report, 55 professional football players contracted a total of 84 sports injuries.

The 2020/2021 sporting season showed the highest number of injured players and the highest number of overall injuries. However, in the 2019/2020 season, the average number of days missed by a player due to injury was higher, approaching 20 days of absence. In the sum of the three seasons, on average, each player contracted 0.77 injuries per season (84 injuries/104 players). It is also fundamental to emphasize that almost half of the players (48.1%) who belonged to Club Sport Marítimo’s squads throughout the three seasons studied never contracted sports injuries, while basically, a third of the players (31.7%) contracted only one injury across the study period. Only one out of five players (20.2%) contracted two or more injuries across the three seasons.

Table 3 summarizes the injury occurrence according to age and sectorial position. Players between the ages of 27 and 30 had the most injuries in the 2019/2020 and 2021/2022 seasons. In the 2020/2021 season, almost half of the injuries belonged to players between 23 and 26 years old. Regarding injuries by sectorial position, goalkeepers consistently presented much fewer injuries than other sectors. Moreover, defenders, midfielders, and forwards had a similar injury pattern per season, with the number of injuries ranging between 6 and 11 injuries per season per sectorial position (Table 3).

Figure 1, Figure 2 and Figure 3 show the injuries’ type, area, and location. Over the three sporting seasons under analysis, the lower limbs were massively affected by injuries, both in the number of injuries and the total days of recovery. The most recurrent injuries were muscular injuries in the hamstrings and quadriceps. Sprains affecting the tibiotarsal structure were also very common and occurred with the highest volume throughout the three seasons.

Table 4 summarizes the data corresponding to the mechanism and severity of injuries. The mechanism of injuries was found to be one of the topics with higher variability. In the 2019/2020 season, most injuries occurred from overload, while in the 2020/2021 season, there was a noticeable balance between overload and traumatic injuries, and in the 2021/2022 season, mainly traumatic injuries occurred. In terms of injury severity, in the three seasons analyzed, moderate injuries (8 to 28 days) were the most frequent in this professional football club. The injuries that caused the most days of abstention during the three seasons were three. Interestingly, all of them required surgical intervention.

Overall, most of the injuries occurred in training situations. However, the incidence of injuries was considerably higher in matches since the exposure time in training was substantially higher than in matches in all the seasons (Table 5).

As expected, most sports injuries that occurred over the study time were non-recurrent. In terms of laterality, there was a balance between the injuries arising in the dominant and non-dominant limbs over the 2019/2020 and 2020/2021 seasons. However, an apparent increase in the injuries affecting the dominant limb and a decrease in the injuries affecting the non-dominant limb was noted in the 2021/2022 season (Table 6).

Figure 4 shows the times in the sports seasons when injuries more frequently occurred. In the 2019/2020 season, the first month of training (July), corresponding to the pre-season period, was the one that presented the highest frequency in terms of sports injuries. The 2020/2021 season had the highest number of injuries in January, approximately at the half of the national championship. In 2021/2022, the season with the fewest sports injuries, the two most frequent times of injury were the months of September and December.

## 4. Discussion

This study aimed to describe and characterize the sports injuries of a professional football team of the First Portuguese League over three consecutive seasons, considering variables such as age, sectorial position, zone, type, specific location, mechanism, severity, occurrence, exposure, incidence, recurrence, and laterality. In this study, muscle injuries and sprains in the lower limbs were the most recurrent, mainly affecting the quadriceps, hamstrings, adductors, and the tibiotarsal structure. The injury incidence was higher in matches, and the months of July and January had the highest injury peaks over the three sports seasons analyzed.

The team average showed 0.77 injuries per athlete over the three seasons. A previous study equally carried out across three seasons reported a higher injury frequency (1.5 injuries per player) [3] than the one found in our research. The differences in the number of injuries per player and team may be related to the competitive level, the training, and the playing conditions each team presents. Indeed, the internal and external training and match loads are variables that are directly related to sports injuries and vary according to the context and competitive level.

Meanwhile, sports injuries led to a player missing 8 to 28 days of training and competition, with an average of 16.6 missed days per injured player. The average number of days missed by an injured player is significantly higher than that reported in the literature, where it is around 7 to 8 absent days [3,17,25]. Additionally, our results suggest that injuries in specific locations such as the knee, which are more severe, will demand a longer period of absence time. A study in Germany’s highest football league for two seasons reinforced this conclusion, stating that the body region most frequently affected by severe injuries, meaning those that result in a longer period of absence, was the knee [23]. Thus, players, coaches, and their staff, as well as the remaining sports agents, should be aware of the need for frequent medical checkups, physical testing, and strengthening all injury prevention and rehabilitation programs to diminish the injuries’ detrimental effects.

The age range of 27 to 30 years was the one more affected by injury. According to another study, players under 26 years had a reduced incidence of injuries compared with their older peers [25]. Other investigations that reached similar results found that older players are more likely to sustain injuries throughout a sports season [15]. This result may be related to several factors, among which include the number of recurrent injuries in a given body zone and location and the load and intensity that an older player has been exposed to over his many years of career, compared with younger players who started their careers recently. Our study sample did not include a sizable number of participants of advanced ages. Only 7% were 30 years or older, making our group of 27 to 30 years players (34%) practically the oldest core of the three seasons analyzed, which is in accordance with the literature that reinforces the interrelationship between advanced age and increased likelihood of injury risk.

The defensive and forward sectorial positions were the most affected by injuries over the three seasons. These results align with those of the previous studies, indicating that defenders and forwards usually have the highest incidence of injury throughout the season [3,17,18,31]. In fact, fullbacks and forwards have higher injury rates than other positions on the field because they must run longer distances with high intensity [38,39]. In contrast, the goalkeepers were consistently the least damaged by injuries throughout the seasons under study. Such data converge with the conclusions of other studies that reported that players playing in this position have a lower risk of injury than their teammates [3,17,31]. According to the literature, the lower number of injuries in goalkeepers is justified by the lower physical demands to which they are subjected and their lower exposure to situations of physical contact [3,31].

As already mentioned, the highest number of injuries predominantly affected the lower limbs, mainly due to muscle injuries in the quadriceps, hamstrings, adductors, and tibiotarsal sprains. These results are supported by the literature, reinforcing that typically more than 80% of football injuries affect the lower limbs, where two out of three injuries are either muscle injuries or sprains that primarily affect the hamstrings, quadriceps, and the tibiotarsal structure [3,14,15,16,17,18,19,23,25,31]. Indeed, due to the tactical–technical actions required in football, there is a predominant solicitation of the lower limbs, justifying their greater exposure to injury [26].

Compared with professional football, several studies point out that basketball [40,41,42,43,44], handball [45,46,47,48,49,50,51], and volleyball [52,53,54,55,56,57,58] are team sports in which the area with the highest frequency of sports injuries is also the lower limbs, with the thigh, ankle, and knee joints having the highest prevalence of injuries. Nevertheless, as in these three sports, the upper limbs handle the object of play, and injuries to the shoulder structure are also recurrent [59,60]. It is interesting that in terms of individual sports, taking the example of court tennis, the lower limbs are also the area of the body with the highest frequency of injuries, mainly the ankles, wrist, knee, foot, and only then the shoulders [61,62,63,64]. Regarding combat sports, using the example of taekwondo, karate, and judo, lower-limb injuries are very common, since kicking the opponent’s torso or head is allowed. In boxing, such injuries are extremely rare. On the other hand, while head injuries are rare in judo, where head contact is prohibited, they are common in karate, boxing, kickboxing, and mixed martial arts, where head contact is allowed. Upper-limb injuries occur most frequently in taekwondo and aikido [65,66,67,68,69].

This study had an evident balance between traumatic and overload injuries. The data presented in the previous injury-related literature support this statement. One study reinforced that traumatic injuries were slightly more prevalent than overload injuries [3]. In contrast, another investigation found that two out of three injuries were from overload [21]. Since there is a direct link between the training load and injury incidence, it is crucial to emphasize the significance of organizing training cycles following the players’ characteristics and physical conditions. This process happens more reliably and consistently when the individual training loads are measured using proper tools. Coaches, players, and technical staff increasingly monitor and analyze the sport’s load using scientific methods [31]. Keeping an eye on the training process is essential to interpret the players’ tiredness indicators. It may help lower the risk of injury. Injury prevention procedures should consider both overload and traumatic injuries since football is a physically demanding and intense sport.

Although most injuries occurred in training situations, the incidence of injury was predominantly higher in matches. These results are consistent with the findings of previous authors [3,18,20,21,22,23,31], reinforcing the importance of monitoring the external and internal load during competition. Consequently, the training content may be adjusted to enhance player performance and preparedness, reducing the likelihood that they will sustain injuries during matches.

The injury recurrence rate in this study is consistent with rates reported in the literature, which range from 8% to 22% [19,24,27]. According to earlier research, these percentage discrepancies may result from the resources available in the respective clinical departments, the club’s infrastructure, and material resources’ capabilities to respond quickly to maximize injury prevention and minimize recovery.

One of the differentiating aspects of our research is the analysis of the most predominant moments, throughout the three seasons, for a peak of injuries to occur in a professional football team. The noticeably higher moments of injuries were the months of July and January. July is the pre-season month in which the intensity and volume of training are higher than during the rest of the season. During this period, there is an increased focus on players’ physical preparation. Since this is a differentiated period from the rest of the sports season, it seems reasonable that there is a higher frequency of injuries at this stage. January is characterized by being approximately the middle moment of the sports season. The athletes already present high levels of fatigue and tiredness, associated with high game intensity and equally high training volume. The volume of injuries in this sportive moment may be related to these crucial aspects. Thus, it is essential to monitor the impact of these phases during the season on the development and enhancement of athletes’ capabilities. Giving tools to all staff members to balance the intensity, volume, and load of training and match sessions is crucial for suitable prevention and recovery programs for sports injuries.

The main limitation of this study is that it did not consider the players’ injury history. However, this investigation brings important theoretical and practical implications for those involved in the professional football context. The characterization of sports injuries is a rising topic that has proven to be a benefit in injury prevention and recovery, as much as in monitoring the training and match situations. It will also help technical staff to support their clinical and sports decisions wisely. The bottom line is that this knowledge will benefit and improve injury prevention and treatment and, consequently, the team’s performance [14,17].

## 5. Conclusions

The main results of this study showed that a higher number of injuries occurred in players with advanced age and mainly affected the lower limbs, particularly through muscle injuries in the quadriceps, hamstrings, adductor muscles, and tibiotarsal sprains. Most of the injuries contracted were classified as moderate, leading the athletes to be absent between 8 and 28 days. The players were more exposed to injuries in match situations than in training situations. In addition, the two times of the season that proved most conducive to injuries were the months of July and January. In summary, our results emphasize the importance of monitoring sports performance, including the occurrence of injuries. This monitoring will assist in identifying the predictive factors of injuries in professional football. Strategies such as designing individualized training programs and optimizing prevention and recovery protocols are crucial to reducing injury incidence.

## Figures and Tables

**Figure 1 ijerph-19-12582-f001:**
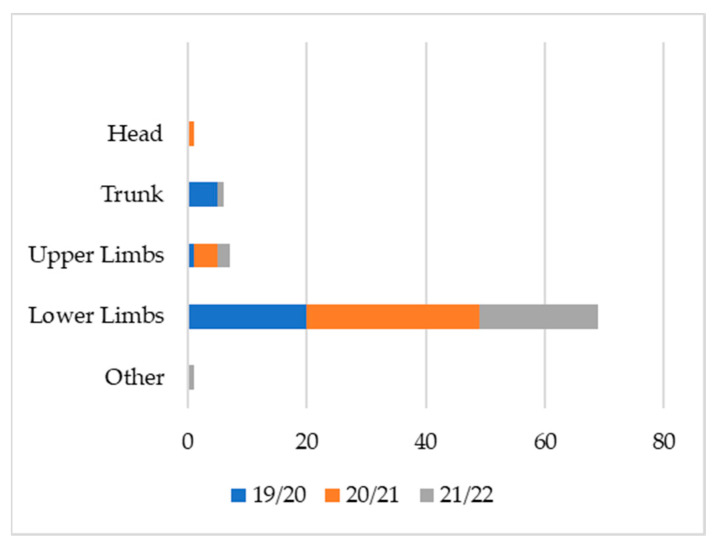
Injury frequency by zone (n).

**Figure 2 ijerph-19-12582-f002:**
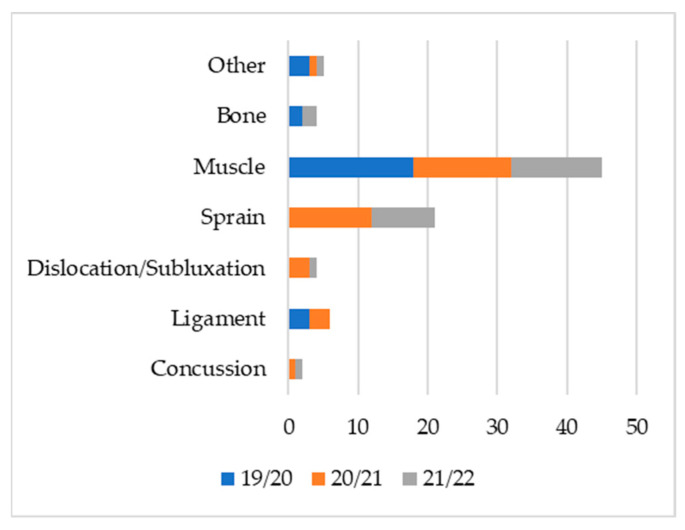
Injury frequency by type (n).

**Figure 3 ijerph-19-12582-f003:**
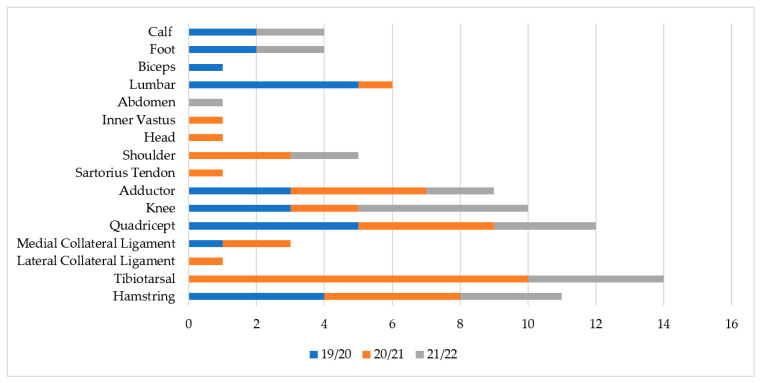
Injury frequency by specific location (n).

**Figure 4 ijerph-19-12582-f004:**
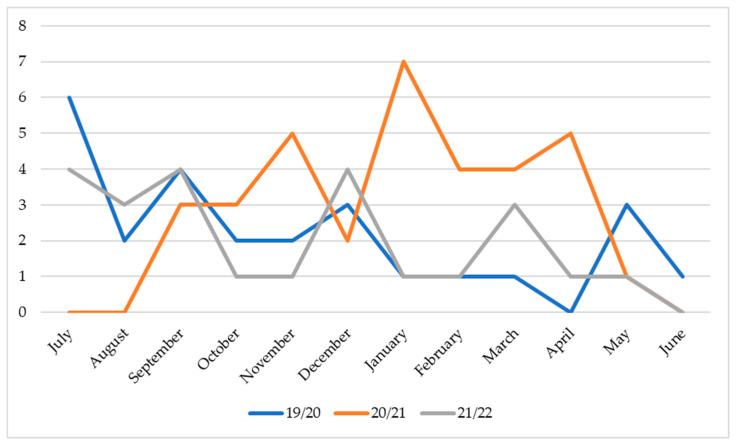
Frequency of injuries per month (n).

**Table 1 ijerph-19-12582-t001:** Characteristics of the team competitive pathway during three sporting seasons.

	2019/2020	2020/2021	2021/2022
Division	1st	1st	1st
League Final Rank	11°	15°	10°
League Points	39	35	38
Portuguese Cup	3rd round	Quarter-Finals	3rd round
Allianz Cup	Group stage	No participation	1st round
Games Played (n)	39	38	36

**Table 2 ijerph-19-12582-t002:** Injuries reported during three consecutive sporting seasons.

Injury Report	2019/2020	2020/2021	2021/2022
Team (No.)	35	36	33
Injured players	17	23	15
Total number of injuries	26	34	24
Average days missed per injury	19.9	14.3	15.6
Average number of injuries per player	0.7	0.9	0.7
Players with 2 or more injuries (No.)	6	9	6

**Table 3 ijerph-19-12582-t003:** Injury occurrence according to age and sectorial position during three consecutive sporting seasons.

	Injury Occurrence *		
	2019/2020	2020/2021	2021/2022
*Age range (years)*			
≤22	5 (19.2%)	6 (17.6%)	3 (12.5%)
23–26	6 (23.1%)	16 (47.1%)	6 (25%)
27–30	12 (46.1%)	10 (29.4%)	10 (41.6%)
31–34	3 (11.6%)	2 (5.9%)	1 (4.2%)
≥35	0 (0%)	0 (0%)	4 (16.7%)
*Sectorial position*			
Goalkeeper	1 (3.9%)	5 (14.7%)	1 (4.2%)
Defender	8 (30.7%)	11 (32.4%)	6 (25.0%)
Midfielder	10 (38.5%)	7 (20.5%)	6 (25.0%)
Forward	7 (26.9%)	11 (32.4%)	11 (45.8%)

* Injury occurrence is presented by the number of injuries and their respective percentage.

**Table 4 ijerph-19-12582-t004:** Injury mechanism and severity during three consecutive sporting seasons.

	2019/2020	2020/2021	2021/2022
*Mechanism*			
Traumatic	8 (30.8%)	17 (50%)	14 (58.4%)
Overload	18 (69.2%)	17 (50%)	8 (33.3%)
Other	0 (0%)	0 (0%)	2 (8.3%)
*Severity* *			
Minimal (1–3 days)	2 (7.8%)	4 (11.7%)	4 (16.7%)
Mild (4–7 days)	5 (19.3%)	7 (20.5%)	7 (29.1%)
Moderate (8–28 days)	18 (69.3%)	17 (50%)	9 (37.5%)
Severe (+28 days)	1 (3.6%)	6 (17.6%)	4 (16.7%)

* Number of days missed by a player due to a sports injury contracted in training or match.

**Table 5 ijerph-19-12582-t005:** Injury occurrence, exposure, and incidence during three consecutive sporting seasons.

	2019/2020	2020/2021	2021/2022
*Occurrence*			
Training	17	20	17
Match	9	14	7
*Exposure (h)*			
Training	7179.5	7154.8	7780
Match	642.1	639.5	650
*Incidence* *			
Training	2.4	2.8	2.2
Match	14	21.9	10.8

* Injury incidence (per 1000 h), h (hours).

**Table 6 ijerph-19-12582-t006:** Injury recurrence and laterality during three consecutive sporting seasons.

	2019/2020	2020/2021	2021/2022
*Recurrence*			
Yes	3 (11.5%)	4 (11.8%)	3 (12.5%)
No	23 (88.5%)	30 (88.2%)	21 (87.5%)
*Laterality*			
Dominant	10 (38.5%)	18 (52.9%)	16 (66.7%)
Non-dominant	11 (42.3%)	16 (47.1%)	6 (25%)
Central	5 (19.2%)	0 (0%)	2 (8.3%)

## Data Availability

The data presented in this study are available upon request from the corresponding author.

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
