# Peer review of "Sports Injuries of a Portuguese Professional Football Team during Three Consecutive Seasons"

_ijerph, 2022, doi:10.3390/ijerph191912582_

Round 1

Reviewer 1 Report

The main aim of this study was to characterize the injuries of a professional football team in the First Portuguese League over three consecutive sports seasons. Regarding the authors, I would like to congratulate and thank them for their effort and motivation involved in this research study. The submitted work is interesting and has been efficiently written. However, I do have a few comments that need to be answered, as well as a few thoughts that I think would enhance this article:

1) What is missing from the introduction is an overview of recent research in the field of football injuries and information on which areas such research is being conducted. It would also be of great value to the article if the introduction could be expanded to include information on whether the authors have already diagnosed similar studies, for example in other countries. Some similar studies are cited in the discussion, but it would be useful to make an introduction to them in the “Introduction” section.

2) What are the research hypotheses? Did the study confirm them? Were these results expected? This should be further elaborated at the end of the introduction.

3) The article states that the study was conducted in accordance with the Declaration of Helsinki, what is extremely important. However, there is no information whether the participants were treated ethically according to the American Psychological Association code of ethics? Please complete this information in the manuscript.

4) The study lacks a statistically in-depth methodology, and the methodology itself is the weakest link in a really interestingly written article. The current statements present only figures and percentages and have not been recalculated in any way to find statistically significant relationships. I do not want to impose specific statistical tests here, but the authors could use the support of an effective statistician and include the analyzed various variables.

5) The bibliography is somewhat meagre and it would have been a good idea to include a few more bibliographic footnotes. In addition, all references should be brought into line with the MDPI Instruction for Authors, which is included on the journal’s website (in particular, bring the cited items into line with MDPI and ACS Style).

Supplementing the manuscript with the above-mentioned scope will in my opinion make a chance for publication in International Journal of Environmental Research and Public Health. I keep my fingers crossed for the final success of the publication. 

Reviewer 2 Report

At the outset, I would like to thank you for the opportunity to review your work.
I consider the topic to be a good one, especially from the perspective of the First Portuguese League sample selection.
In order for the text to fully meet the requirements of this journal, I propose the following changes:

1. Subsections 2.3. and 2.4. please merge into one "Procedure". 2. Presentation of the results is based on statistical averages. It should be expanded with statistically significant differences between the means.
3. In order to fully justify the results, they should be referred in the discussion to other team sports and then to individual sports and combat sports.
4. Please expand the literature with topics adequate to the guidance from ad.3

Reviewer 3 Report

Dear, authors thanks for the submission. 

Congratulations for your ability to follow a professional team for 3 years and the quality of the manuscript. I have just some minor comments that need to be addressed. Well done!

Round 2

Reviewer 1 Report

I would like to thank the authors of the article for including most of my suggestions. The manuscript has been significantly strengthened and I have no further comments.

Reviewer 2 Report

I recommend the work for publication.